# Planning Implementation Success of Syncope Clinical Practice Guidelines in the Emergency Department Using CFIR Framework

**DOI:** 10.3390/medicina57060570

**Published:** 2021-06-03

**Authors:** Jing Li, Susan S. Smyth, Jessica M. Clouser, Colleen A. McMullen, Vedant Gupta, Mark V. Williams

**Affiliations:** 1Center for Health Services Research, University of Kentucky, Waller Health Care Annex, 304A, Lexington, KY 40536, USA; jess.clouser@uky.edu (J.M.C.); mark.will@uky.edu (M.V.W.); 2Department of Cardiovascular Medicine, Gill Heart & Vascular Institute, University of Kentucky, 900 S. Limestone St., CTW320, Lexington, KY 40536, USA; susansmyth@uky.edu (S.S.S.); cmcmu2@uky.edu (C.A.M.); vedant.gupta@uky.edu (V.G.)

**Keywords:** syncope, emergency department, diagnosis, risk stratification

## Abstract

*Background and Objectives:* Overuse and inappropriate use of testing and hospital admission are common in syncope evaluation and management. Though guidelines are available to optimize syncope care, research indicates that current clinical guidelines have not significantly impacted resource utilization surrounding emergency department (ED) evaluation of syncope. Matching implementation strategies to barriers and facilitators and tailoring strategies to local context hold significant promise for a successful implementation of clinical practice guidelines (CPG). Our team applied implementation science principles to develop a stakeholder-based implementation strategy. *Methods and Materials:* We partnered with patients, family caregivers, frontline clinicians and staff, and health system administrators at four health systems to conduct quantitative surveys and qualitative interviews for context assessment. The identification of implementation strategies was done by applying the CFIR-ERIC Implementation Strategy Matching Tool and soliciting stakeholders’ inputs. We then co-designed with patients and frontline teams, and developed and tested specific strategies. *Results:* A total of 114 clinicians completed surveys and 32 clinicians and stakeholders participated in interviews. Results from the surveys and interviews indicated low awareness of syncope guidelines, communication challenges with patients, lack of CPG protocol integration into ED workflows, and organizational process to change as major barriers to CPG implementation. Thirty-one patients and their family caregivers participated in interviews and expressed their expectations: clarity regarding their diagnosis, context surrounding care plan and diagnostic testing, and a desire to feel cared about. Identifying change methods to address the clinician barriers and patients and family caregivers expectations informed development of the multilevel, multicomponent implementation strategy, MISSION, which includes patient educational materials, mentored implementation, academic detailing, Syncope Optimal Care Pathway and a corresponding mobile app, and Lean quality improvement methods. The pilot of MISSION demonstrated feasibility, acceptability and initial success on appropriate testing. *Conclusions:* Effective multifaceted implementation strategies that target individuals, teams, and healthcare systems can be employed to plan successful implementation and promote adherence to syncope CPGs.

## 1. Introduction

Syncope is a common yet complex presenting symptom and requires thoughtful and efficient evaluation to determine its etiology. Estimates indicate that one-half of all Americans will experience loss of consciousness during their lives, with recurrence rates as high as 13.5% [1]. The incidence of syncope is roughly bimodal, with a peak in late adolescence to early adulthood, typically vasovagal in origin [2], and a second peak in older age, with a sharp rise after age 70 years [3]. Approximately 1% to 3% of all emergency department (ED) visits, as many as atrial fibrillation, and up to 6% of all hospital admissions are due to syncope [1,4,5]. Though vasovagal reflex-mediated syncope and orthostatic hypotension are the two most common types with benign courses [6], a cardiac etiology of syncope is associated with significantly higher rates of morbidity and mortality [3].

Patients who present to the ED tend to be older and are more likely to have a cardiac etiology [7]. Notably, experiencing syncope affects patients’ quality of life (QoL), and those with more frequent syncope report overall lower physical and mental health and impairment in activities of daily living [8,9,10,11,12,13]. The QoL among patients with recurrent syncope appears equivalent to those with severe rheumatoid arthritis or chronic lower back pain [10]. Recurrent syncope can also lead to long-term facility stay and a devastating loss of independence [14]. In addition to the negative effects on QoL, syncope also has an economic impact. The U.S. Healthcare Utilization Project has estimated total annual hospital costs of greater than $4.1 billion in 2014 dollars with a mean cost of $9400 per admission [15]. One 2017 article showed that, after adjusting for inflation, the median hospital charge for a single admission for syncope increased by 1.5 times from the preceding decade [16].

Due to concerns that patients presenting with syncope are at risk for an impending catastrophic event, overuse and inappropriate use of testing and hospital admission are common [17,18,19,20]. Indisputably, among patients who present with syncope, clinicians must identify those at high risk of adverse outcomes. Nonetheless, the majority are at low risk. To assist clinicians in assessing patient risk, several syncope risk stratification calculators have been developed over the last 20 years; however, one study found that the concordance between different risk scores was only moderate and the application of both decision rules and clinical judgement may lead to some clinical benefit [21]. A body of literature documents under-utilization of efficient tests, over-utilization of unnecessary tests, excess rates of admissions with limited diagnostic or therapeutic yield, over-expenditure associated with syncope management, and heightened risk to patients due to unnecessary tests and hospitalizations, including iatrogenic harms such as medication errors and in-hospital delirium [17,18,19,22]. Given the frequency of syncope as a symptom, the cumulative cost and burden to the healthcare system and patients is substantial. 

Aiming to provide guidance on optimizing the evaluation and management of syncope, a collaboration of the American College of Emergency Physicians, Society for Academic Emergency Medicine, American College of Cardiology (ACC), American Heart Association (AHA) and Heart Rhythm Society (HRS) issued a Guideline for the Evaluation and Management of Patients With Syncope in 2017 [15]. The 2017 Syncope Guideline represents an effort to standardize clinical practice and reduce unnecessary services. However, the mere existence of a guideline does not guarantee effective use. Evidence shows that the development of clinical guidelines alone is often not sufficient, even if recommendations in the guideline have been demonstrated to be effective on the structure, process and/or outcomes of patient care [23,24,25,26,27]. Indeed, one recent study suggested that the current clinical guidelines have not significantly impacted resource utilization surrounding ED evaluation of syncope, and novel strategies are keenly needed to change ED practice patterns for such patients [28]. Matching implementation strategies to barriers and facilitators for the use of the syncope guideline and tailoring strategies to local context hold significant promise for a successful implementation [29,30,31]. However, evidence on effective implementation strategies for syncope care in the ED is scarce. Project MISSION, leveraging an engaged interdisciplinary team, aimed to facilitate the efficient and systematic implementation of high-value care to patients presenting to an ED with syncope. Our study team applied implementation science to develop and test a stakeholder-based implementation strategy, MISSION (Multicomponent, Multilevel Implementation Strategy for Syncope Optimal Care Through Engagement).

## 2. Materials and Methods

### 2.1. Study Setting and Participants

To maximize the probability that the implementation strategy will be valuable for widespread adoption and scale-up in different ED settings, Project MISSION included a diverse group of health systems and hospitals: an academic medical center (AMC); an urban faith-based community health system; a not-for-profit health system serving a predominantly rural Appalachian population; and a community teaching hospital in a suburb. At each facility, the target participants included emergency medicine (EM), hospital medicine (HM), and cardiology clinicians and stakeholders (e.g., primary care provider, nurse manager, diagnostic test/procedure manager). Patients and family caregivers were recruited for interview from the AMC.

### 2.2. Study Framework

The Consolidated Framework of Implementation Research (CFIR) [32] is commonly used to guide the design, implementation and evaluation of strategies. CFIR was deemed to meet the needs of our project, as a determinant framework that can be used to identify determinants (i.e., barriers and facilitators) thought to affect the likelihood of a clinical guideline being translated into routine care and influence the implementation process at different levels (from the user to the program provider, to the organizational level) [33]. In Figure 1, we delineate the possible influencing factors for syncope clinical practice guideline (CPG) adoption and implementation through multi-stakeholder effort. The effectiveness of a multicomponent, multilevel implementation strategy will be mediated by the fidelity with which the intervention is delivered, and patient outcomes will be moderated by several patient-level factors. The entire process occurs within a context composed of system-level, organizational-level, and provider-level factors known to influence the implementation of a CPG [34].

### 2.3. Development of Implementation Strategy (MISSION)

Table 1 delineates the influencing factors for syncope CPG adoption and implementation and lists the activities performed to assess determinants. The study team partnered with patients, family caregivers, frontline clinicians and staff, and administrators to assess contextual factors (e.g., patient preferences and needs, clinician perceptions, local organizational structure, operating philosophy and culture) and readiness for syncope guideline implementation. We conducted focus groups and interviews of patients and their family caregivers, clinicians and staff, and administrators [35,36]. We also surveyed clinicians and staff to understand unique challenges and barriers in each of these systems. The implementation questions addressed: (1) what are the facilitators/barriers to delivering guideline-based evaluation and management of syncope within the local context, (2) how likely will the recommendations be delivered as prescribed (fidelity), and (3) what strategies might maximize the facilitators and overcome barriers to implementation? After completing context assessments, identifying barriers and facilitators, and soliciting stakeholders’ inputs on strategies, we used the CFIR-ERIC (the Expert Recommendations for Implementing Change compilation) Implementation Strategy Matching Tool [37] to help select and tailor MISSION components to mitigate barriers and leverage facilitators.

### 2.4. Analysis

Descriptive statistics were calculated for each survey item. Bivariate analyses were used to assess associations between characteristics (clinician specialty, hospital setting) and attitudes and readiness among respondents. Data were analyzed using SAS 9.4 (SAS Institute, Cary, NC, USA).

The interviews were transcribed verbatim for content analysis. The study team developed initial code books based on their clinical and implementation expertise. Coding took place in two stages. During the first stage, two research staff coders independently reviewed the transcripts to identify unique themes using NVivo 12 software (QSR International, Melbourne, Australia). After the first round of coding, both coders met to discuss any disagreements and refine the schema of codes and to refine the codebook for additional rounds of coding. Then, two coders met with the study team’s qualitative expert to discuss and refine the coding schema by merging, reformulating, or rephrasing codes to more accurately fit the data and create one cohesive codebook. The two original coders then co-coded each transcript. Analytical memos were created and discussed as a group over a series of weekly meetings with the goal of refining and finalizing themes and categories.

This study was approved by the University of Kentucky Institutional Review Board (protocol #45255). 

## 3. Results

### 3.1. Clinician Survey and Interview

Project MISSION achieved broad engagement across multiple practice settings. One hundred fourteen clinicians completed surveys and thirty-two clinicians and stakeholders participated in interviews [22,35]. The survey and interview results have been reported in detail elsewhere [22,35]. Briefly, among clinicians, awareness and implementation of the 2017 Syncope Guideline was low. We identified practice gaps in under-reporting of orthostatic vital signs and overuse of cardiac and neurologic imaging, as well as barriers to adoption and implementation of evidence-based care across multiple levels. Survey results revealed that overall attitude toward evidence-based practices was moderate, and implementation of new guidelines were seen as a burden, potentially decreasing the likelihood of compliance. Of the multiple patient, provider, and organization-related barriers to syncope guideline implementation, we identified communication challenges with patients, lack of CPG protocol integration into ED workflows, and organizational process to change as major barriers to implement CPGs in syncope care [35].

### 3.2. Patient and Family Caregiver Focus Group

Project MISSION focus group sessions were conducted to understand patient needs, values and preferences. A total of 31 patients and their family caregivers, 23 patients and 8 caregivers, participated in interviews [36]. They described their expectations when presenting to the ED with syncope including: (1) clarity regarding their diagnosis or cause of their syncope, (2) context surrounding care plan and care teams’ approach to diagnostic testing, and (3) desire to feel seen, heard and cared about by the healthcare team. 

### 3.3. Implementation Barrier—Strategy Mapping

The findings from quantitative surveys and qualitative interviews helped guide decisions about the types of strategies that may be appropriate and match the needs of the local context. Based on the CFIR-ERIC Implementation Strategy Matching Tool (www.cfirguide.org (accessed on 20 March 2021)), we elicited input from the study team, frontline clinicians and staff, and administrators on choosing which ERIC strategies would best address specific CFIR-based barriers in guideline recommended syncope evaluation and management. Table 2 lists the identified CFIR barriers and ERIC recommended strategies. 

### 3.4. MISSION Implementation Strategy Components

After assessing and understanding determinants within the local context and identifying change methods to address those determinants, the last step was to develop strategy components to address the determinants considering how barriers interact with syncope care-specific needs. This process was also complemented with Fernandez and colleagues’ five-step Implementation Process [38] and iterative feedback from stakeholders to further operationalize these components. Table 3 shows the multicomponent, multilevel implementation strategy (i.e., MISSION) components and expected functions/outcomes achieved.

Syncope patients see testing as a means to achieve clarity on their otherwise ambiguous condition. Clinicians can focus on two-way communication by engaging in active listening, obtaining a complete patient history, and explaining the rationale for or against various testing options. Printed educational materials are one of the most common forms of communicating guidelines. Our team developed educational videos (intake and discharge videos) to help align patient expectations regarding testing to fit with guideline recommendations, as well as tailored patient educational materials to better explain their specific syncope diagnosis. The Hospital Patient Education Department and the Patient and Family Advisory Group reviewed all educational materials, providing feedback and editing the materials to ensure an appropriate reading level. Additionally, we created a discharge document incorporating principles of adult learning theory and health literacy to help providers educate patients on the details of their diagnosis, preventive measures, and instructions to follow at the time of discharge.

In addition to clinical decision support (CDS) tools, the strategies aiming to promote clinician behavior change and optimize clinical process include mentored implementation combined with academic detailing. Mentored implementation provides external expert facilitation to enable and support health systems to make and sustain change, and efficiently integrate efforts into current workflow. It also facilitates active stakeholder engagement, offers ongoing support, and equips local champions for sustainability. This approach is proven to enhance adoption and implementation of evidence-based programs and innovations [40,41]. Academic detailing [42,43,44] is peer-to-peer educational outreach and addresses situations where there is an opportunity to change clinician behavior with focused and practical educational content. It can also help build leadership’s buy-in to the proposed practice changes and help them understand how they can help the frontline implement these changes.

Project MISSION also created implementation strategies that address the process of integrating essential content from syncope CPGs to the local practice context and workflow. Clinical protocols provide specific guidance for management of groups of patients, in an algorithmic structure that facilitates clinical decision-making, tailored to the local environment. With input from diverse health systems and engagement of interdisciplinary expertise, our study team developed the Syncope Optimal Care Protocol based on the 2017 Guideline. The Syncope Optimal Care Protocol provides a standardized clinical pathway that has flexibility to make it more attractive to clinicians and aids in reducing variability, while improving quality and lowering cost. Next, a MISSION mobile application (App) was designed to be a practical tool for the implementation of the Syncope Optimal Care Protocol and serve as a CDS tool for syncope diagnosis and prognosis that walks users through clinical assessment in a clear and concise manner, and provides recommendations based on input from the user [39].

Finally, to address workflow compatibility and care process redesign, Lean quality improvement (QI) [45] tools were selected to be part of the implementation strategies. Lean generally focuses on how a process is currently operating and what opportunities exist to improve the process in a local setting, and therefore is a best practice in tailoring implementation. Application of Lean QI methods and tools aims to increase the likelihood of sustaining the daily practice and maximizing its impact in each health system.

### 3.5. MISSION Implementation Strategy Pilot

The Project MISSION implementation strategy was piloted from 17 Feb through 13 March 2020 at an AMC ED. The pilot stopped earlier than scheduled due to COVID-19, but demonstrated feasibility and acceptability, with 91.7% (22/24) of approached patients watching education videos with voiced approval, and 34 clinicians downloading and using the MISSION App. The 2017 Syncope Guidelines recommends that orthostatic vital signs, a low-cost, effective diagnostic test, are included as a required part of the physical examination for patients presenting with syncope. However, according to recent literature [46] and data reported by hospitals in this study, orthostatic vitals were underused, being performed on only 15% to 40% of patients. Routine head CT scan without a severe coexisting injury or disease is not recommended in the 2017 Syncope Guideline. A 2019 systematic review showed that more than half of patients with syncope underwent head CT scan at ED, but with a diagnostic yield of only 1.1% to 3.8% [47]. Based on the literature and stakeholders’ recommendations, orthostatic vital signs and head CT orders are two major implementation outcome measures in our study. Given the low baseline, a relative 50% increase in orthostatic vital signs will be considered as clinically significant, and given the high baseline, a relative 20% reduction in head CT scan orders will be clinically significant. Following MISSION implementation, we found that orthostatic vital sign measurement increased from 29% to 43% (χ^2^ statistic = 4.2664, *p*-value = 0.0389) and inappropriate head CT orders reduced from 48% to 37% (χ^2^ statistic = 2.3641, *p*-value = 0.1242). This demonstrated a clinically significant improvement in implementing CPGs in the evaluation and management of syncope. 

## 4. Discussion

### 4.1. Evaluation of Barriers Is a Necessity in Planning CPGs Implementation

Despite substantial efforts by medical researchers and professional societies [15,48,49], overuse and inappropriate use of testing and hospital admission are common in patients presenting with syncope. The most efficient solution to improve patient outcomes is most likely to adopt standardized criteria for evaluation and treatment administration based on the recommendations contained in guidelines. However, the uneven implementation of evidence-based CPGs is widely recognized as a continuing challenge to improving healthcare delivery and public health [50,51]. Implementation science provides an empirical base for promoting adoption of CPGs and its research is dedicated to accelerating the pace of implementing evidence-based interventions in real-world healthcare settings. What determines the rate and extent of adoption is the interaction among characteristics of the CPG, the intended users, and a particular context of care setting. As part of the clinical guideline implementation planning process, a more detailed evaluation of underlying barriers and facilitators and how these determinants can be addressed by strategies is needed.

### 4.2. Local Context Tailored Implementation Strategy Is Essential

While tailoring to local context seems intuitive, most studies have not tailored implementation strategies to context. Healthcare delivery settings influence every step of how care is given, yet far more work is needed to effectively describe and link these structural and process characteristics to outcomes and to develop setting-changing interventions to improve care. Numerous conceptual frameworks (e.g., CFIR) have been developed to guide the identification and systematically assess potential determinants within local settings. Project MISSION was the first effort that specifically applied IS principles and methods to develop strategies and plan implementation processes to overcome multilevel barriers to deliver guideline-recommended, high-value care to patients presenting with syncope in the ED. It integrated behavioral interventions and healthcare process redesign, used stakeholder-engaged and local-context congruent approaches, and fostered a learning health system approach spanning an academic medical center and community hospitals. Development of MISSION ensured tailoring of implementation strategies in the local setting to accommodate variations and to sustain improved syncope care through tailored implementation. For example, patient educational videos can be edited by inserting a tailored intro and outro delivered by a recognizable, local clinician to enhance patient buy-in. In addition, the video can be presented in various ways based on each system’s infrastructure: via its system-wide patient education platform (either standalone or part of electronic health record-EHR), through a QR code to play on patients’ smartphones, or through an iPad in patient rooms. Another example, supported by an external implementation mentor, is that the local implementation team can use local detailed process maps to systematically identify process steps with opportunities, and test and refine strategies to increase guideline-recommended syncope care delivery through iterative test cycles. 

Limitations to the project should be mentioned. While the organizational structure, hospital characteristics, and patient populations are diverse, the themes presented in this paper were generated based on the responses of participants located in the same state. Second, pilot data were limited due to COVID-19 research restrictions. A larger pilot and implementation will be launched in our state, as well as others, once hospitals move into post-pandemic operation.

Identifying a barrier is not sufficient to guide the choice of an implementation strategy. The causes of each barrier must be specified along with the desired outcome, and the specific methods or techniques must be identified and operationalized into concrete strategies to influence these determinants. A process akin to systematically identifying barriers, change methods for addressing them, and development or selection for specific strategies is essential in implementing CPGs. Unfortunately, this approach is not typically followed, leading to gaps in understanding which strategies work and why they produce their effects. Project MISSION represents an example that utilized CFIR to characterize contextual determinants of CPG use, analyzed those determinants systematically via a theoretical framework, identified specific behavior change targets, and then selected relevant implementation strategies. Pilot testing of MISSION demonstrated feasibility and acceptability among patients, frontline clinicians and staff, and administrators. Our next step is to determine whether MISSION is an effective, generalizable strategy in a pragmatic clinical trial across multiple health systems.

## 5. Conclusions

Effective multifaceted implementation strategies targeting individuals, teams, and healthcare systems should be employed to plan successful implementation and promote adherence to CPGs. MISSION, developed by following implementation science principles, can optimize syncope care and translate CPGs into widespread clinical practice. 

## Figures and Tables

**Figure 1 medicina-57-00570-f001:**
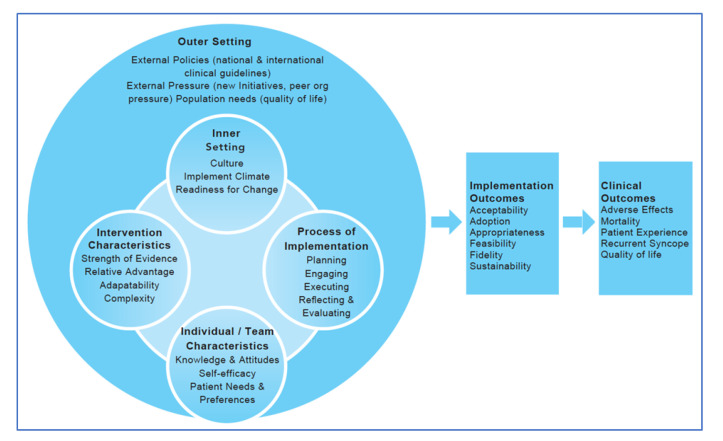
Project MISSION Guiding Framework, adapted from CFIR [32].

**Table 1 medicina-57-00570-t001:** Study activities to assess barriers and facilitators.

Domain	Construct	Assessment
Inner Setting	Readiness for implementation	1. Survey—Organizational Readiness to Change Assessment (ORCA) 2. Focus groups and key informant interviews—clinicians and stakeholders
	Structural characteristics (e.g., availability of electronic information infrastructure)	Focus groups and key informant interviews—clinicians and stakeholders
Individual Characteristics	Patient needs, values, and preferences	Focus groups—patients and family caregivers
	Provider attitudes to evidence-based practices	Survey—revised Evidence-Based Practice Attitudes Scale (EBPAS-36)
Intervention Characteristics	Strength of evidence, relative advantage, adaptability, and complexity	Focus groups and key informant interviews—clinicians and stakeholders

**Table 2 medicina-57-00570-t002:** Syncope Clinical Practice Guideline (CPG) implementation barriers and recommended strategies.

Identified CFIR Barriers	ERIC-Endorsed, MISSION Stakeholder-Recommended Strategies
Intervention—Complexity	Promote adaptability Develop an implementation toolkit Conduct cyclical small tests of change Conduct ongoing training
Clinicians and stakeholders believe that the syncope CPG is complex based on their perception of duration, scope, disruptiveness, and number of steps needed to implement.	
Outer Setting—Patient Needs	Prepare patients to be active participants Involve patients and family caregivers Equip clinicians with tools to help communication
Clinicians feel the pressure to satisfy patients (i.e., consumerism). Patient needs are not known or fully understood by clinicians.	
Inner Setting—Culture and Learning Climate	Facilitation by external agent/adviser Identify and prepare champions Identify and prepare core implementation team Recruit, designate and train for leadership Conduct local consensus discussions Organize clinician implementation team meetings
Cultural norms and basic assumptions hinder implementation. Clinicians do not feel that they are essential, valued, and knowledgeable partners in the implementation process. Clinicians do not feel psychologically safe to implement guidelines.	
Inner Setting—Compatibility	Conduct local consensus discussions Promote adaptability Tailor strategies Lean QI methods
The syncope CPG recommendations do not fit well with existing workflows, nor align well with clinicians’ own needs.	
Individuals—Knowledge & Beliefs about the Intervention	Conduct educational meetings Develop educational materials Conduct educational outreach visits Identify and prepare champions Inform local opinion leaders
Clinicians are not familiar with 2017 Syncope Guideline. Some clinicians have negative attitudes toward guidelines and place low value on implementing them.	
Individuals—Self-efficacy	Identify and prepare champions Provide ongoing consultation Conduct ongoing training Make training dynamic
Clinicians and stakeholders do not have confidence in their capabilities to execute courses of action to achieve implementation goals.	

**Table 3 medicina-57-00570-t003:** MISSION components.

MISSION Components	Expected Functions/Outcomes
Patient educational materials	Prepare patients and family caregivers Assist clinician with challenging communications
Video: Setting Expectations; What’s Next? Syncope Types: one-page document facilitating clinician-patient communication	
External implementation mentor	Create or Enhance culture of learning health systems and continuous improvement Enhanced leadership engagement in and endorsement of CPG implementation in syncope care Enhanced self-efficacy of local implementation team Knowledge and skill transfer to local team and local implementation capacity building
Pre-implementation planning visit Series of ten monthly virtual meetings with local implementation team, including champion, implementation leader and opinion leaders Mid-implementation visit Technical assistance with Lean QI methods	
Academic detailing	Clinician attitude and behavior changes Adherence to syncope CPGs and improvements in patient outcomes
Direct educational outreach to local clinicians Clinical vignettes Discussion with clinicians in their practice setting	
Syncope Optimal Care Protocol	Frontline-endorsed protocol as institutional policy Enhanced clinician receptivity to standardized clinical pathway with flexibility
Syncope MISSION App [39] (iOS and Android)	Operationalized Syncope Optimal Care Protocol Enhanced clinical decision support
Lean QI methods	Redesigned/optimized care process/workflow at ED with syncope CPGs integrated
Syncope MISSION Implementation Tool	Operationalized implementation processes

## Data Availability

Data available from the authors upon reasonable request.

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
