# Peer review of "Planning Implementation Success of Syncope Clinical Practice Guidelines in the Emergency Department Using CFIR Framework"

_medicina, 2021, doi:10.3390/medicina57060570_

Round 1

Reviewer 1 Report

Authors have illustrated implementation of Syncope management guideline which is critical to the field in order to reduce health care burden and prioritize treatments. The manuscript is thorough, focused and well written.

  1. It will be great if the authors can divide the 'Result' and 'Discussion' sections in subsections with subtitles. It will further enhance the readability of the article.
  2. It will be great if the manuscript is edited for typos.

Reviewer 2 Report

The study “Planning Implementation Success of Syncope Clinical Practice Guidelines in the Emergency Department using CFIR Frame-work” by Li and colleagues, addresses the problem of the strategies to be developed for a successful implementation of Clinical Practice Guideline (CPG) on syncope, given that the current clinical guidelines have not significantly impacted resource utilization related to the syncope evaluation in the Emergency Department clinical setting. The Authors set a stakeholder-based implementation strategy called MISSION (Multicomponent Implementation Strategy for Syncope Optimal care through eNgagement), following implementation science (IS) strategies and guidelines. An interdisciplinary team was created comprising patients, family caregivers, ED clinicians and staff and health system administrators at four health systems to conduct quantitative surveys and interviews.

Authors identified change methods addressing existing barriers and set up adequate correcting actions that were providing all the above figures with educational materials, set mentored implementation stages, definition of the Syncope Optimal Care Pathway, building up a Mobile App and disseminating lean quality improvement methods.  

They concluded that multifaceted strategies targeting individuals, teams and health care systems may be effective in promoting adherence to syncope CPGs.

Few minor issues ought to be addressed to improve the excellent quality of the current study, as follows:

  1. This reviewer is aware of the possible delays in the Mission project development because of the COVID-19 outbreak. Nonetheless, it would be important for the readers to know in more details your strategies to assess the expected changes in functions and outcomes previously stated. How would you evaluate the effectiveness of your strategies? What would be your bench mark? Some examples and comments would be wise.
  2. A pivotal study dealing with syncope management in the ED (Am J Emerg Med. 2010 May;28(4):432-9.  doi: 10.1016/j.ajem.2008.12.039) is missing from the reference list, should be quoted and briefly discussed.
  3. The reference list might take advantage from other quotations to be added. All the following are dealing on syncope management:
  • Brignole M, et al; ESC Scientific Document Group. Practical Instructions for the 2018 ESC Guidelines for the diagnosis and management of syncope. Eur Heart J. 2018 Jun 1;39(21):e43-e80. doi: 10.1093/eurheartj/ehy071

  • Brignole M, et al; ESC Scientific Document Group . 2018 ESC Guidelines for the diagnosis and management of syncope. Eur Heart J. 2018 Jun 1;39(21):1883-1948. doi: 10.1093/eurheartj/ehy037
